# Universal scaling laws of keyhole stability and porosity in 3D printing of metals

Zhengtao Gan [1✉], Orion L. Kafka [1,4], Niranjan Parab [2,5], Cang Zhao [2,6], Lichao Fang [1], Olle Heinonen [3], Tao Sun [2,7] & Wing Kam Liu [1✉]

Metal three-dimensional (3D) printing includes a vast number of operation and material parameters with complex dependencies, which significantly complicates process optimization, materials development, and real-time monitoring and control. We leverage ultrahigh-speed synchrotron X-ray imaging and high-fidelity multiphysics modeling to identify simple yet universal scaling laws for keyhole stability and porosity in metal 3D printing. The laws apply broadly and remain accurate for different materials, processing conditions, and printing machines. We define a dimensionless number, the Keyhole number, to predict aspect ratio of a keyhole and the morphological transition from stable at low Keyhole number to chaotic at high Keyhole number. Furthermore, we discover inherent correlation between keyhole stability and porosity formation in metal 3D printing. By reducing the dimensions of the formulation of these challenging problems, the compact scaling laws will aid process optimization and defect elimination during metal 3D printing, and potentially lead to a quantitative predictive framework.

[1] Department of Mechanical Engineering, Northwestern University, Evanston, IL, USA. [2] X-ray Science Division, Argonne National Laboratory, Lemont, IL, USA. [3] Materials Science Division, Argonne National Laboratory, Lemont, IL, USA. [4] Present address: Applied Chemicals and Materials Division, National Institute of Standards and Technology, Boulder, CO, USA. [5] Present address: Intel Corporation, Hillsboro, OR, USA. [6] Present address: Department of Mechanical Engineering, Tsinghua University, Beijing, China. [7] Present address: Department of Materials Science and Engineering, University of Virginia, Charlottesville, VA, USA. ✉email: zhengtao.gan@northwestern.edu; w-liu@northwestern.edu

In metal three-dimensional (3D) printing, also called additive manufacturing (AM), components are typically built layer by layer via local melting and (re)solidification of feedstock materials, often gas-atomized metallic powders. This process provides considerable freedom to design local features, such as geometry and composition, in addition to enhancing manufacturing flexibility and reducing material waste. However, metal 3D printing has a vast number of parameters with complex interactions and dependencies to be considered when making a component[1]. Many authors have quantified the effects of various individual parameters or groups of parameters[2–6]. However, universal physical relationships, which are proven to be valid for different materials, processing conditions, and machines, have remained elusive. The multivariate and multiphysics nature of the metal 3D printing processes complicates parameter optimization, materials development and selection, and real-time process control.

During laser powder bed fusion 3D printing, a topological depression (termed a keyhole) frequently forms, which is caused by vaporization-induced recoil pressure[7]. Keyhole dynamics is inherently difficult to understand and predict because of its complex dependence upon many physical mechanisms but important to be able to quantify because it is highly related to energy absorption and defect formation in metal 3D printing. The geometry of the keyhole significantly affects the energy coupling mechanisms between the high-power laser and the material[3], which leads to unusual melt pool dynamics[8] and solidification defects[1]. An instable keyhole might also cause severe process instability and structural defects, including porosity, balling effect, spattering, and unusual microstructural phases[9]. Recent research capturing meso-nanosecond keyhole dynamics with high-fidelity simulations discovered keyhole-induced back-spattering and frozen depression defects[10]. Although such keyholes were studied for laser welding in 1970s[11], high-quality in situ experimental data on keyhole dynamics only recently became available via high-speed X-ray imaging. These high-energy X-ray imaging experiments have been conducted in laser melting of bare plate[12], powder bed[13–18], and powder flow[19]. As compared with traditional post-mortem characterization of cross-sectioned fusion regions[3–5], ultrahigh-speed X-ray imaging provides adequate temporal and spatial resolutions for probing keyhole evolution and stability.

Another longstanding issue in metal 3D printing and welding is the generation of excessive porosity. Much effort has been directed at determining the physics underlying this phenomenon, as well as finding ways to eliminate or ameliorate it[20]. Several mechanisms that lead to porosity formation have been identified, such as lack of fusion[21], instability of the depression zone[6], vaporization of volatile elements[22], and hydrogen precipitation[23]. However, these efforts are still far from producing a predictive model for porosity—it is challenging to distinguish the quantitative impact of different mechanisms on the final, observed porosity. This makes it impossible to predict the porosity type and magnitude and to optimize processing conditions to build pore-free parts. Elegant insights into the behavior of complex systems, such as metal 3D printing, can be provided by low-dimensional patterns expressed as compact mathematical equations or scaling laws. This adds simplicity to highly multivariate and/or multi-dimensional systems and helps guide process tailoring toward rapid scientific discovery and optimum engineering design[24].

In this work, in order to identify scaling laws in 3D printing we start by generating and collecting in situ synchrotron X-ray imaging[2] data with various process parameters and materials (Supplementary Data 1). We then apply dimensional analysis[25] to normalized governing equations of the system to identify scaled parameters and achieve dimensionality reduction. A multiphysics model is used to interpret energy absorption mechanisms and support the scaling laws. Those compact scaling laws with the property of dimensional homogeneity can be confirmed via the application of nonlinear symbolic regression method such as genetic programming[26].

## Results

**Universal scaling of keyhole aspect ratio and its stability.** The aspect ratio of the keyhole, $e^*$, defined as keyhole depth $e$ divided by laser spot radius $r_0$ (Supplementary Fig. 1), is a critical parameter in laser manufacturing processes, because it can be used to classify the laser–metal interaction into conduction, transition, and keyhole modes[7]. The keyhole aspect ratio in a process is determined by the process parameters and material characteristics. The keyhole depth is directly quantified from in situ X-ray images in this study, while most of literature measures the depth post facto from cross-sections of the fusion region[3–5]. We found that both the keyhole aspect ratio and its fluctuation exhibit universal scaling with only one dimensionless parameter (Fig. 1a), thus we define this dimensionless parameter as the Keyhole number Ke,

$$\mathrm{Ke} = \frac{\eta P}{(T_1 - T_0)\pi \rho C_p \sqrt{\alpha V_s r_0^3}}, \qquad (1)$$

which is defined by the absorptivity $\eta$, laser power $P$, liquidus temperature $T_1$, substrate temperature $T_0$, density $\rho$, heat capacity $C_p$, thermal diffusivity $\alpha$, scan speed $V_s$, and laser radius $r_0$. Some ancillary factors, such as substrate temperature and environmental pressure, are fixed in all the experiments. Their effects are thus not quantified in this work. The form of Keyhole number can be derived from heat transfer theory with simplifying assumptions (Supplementary Method 3). There is a nearly linear relationship between the keyhole aspect ratio $e^*$ and Ke (shown by the dashed line in Fig. 1a):

$$e^* = 0.4(\mathrm{Ke} - 1.4). \qquad (2)$$

The identified scaling law for keyhole aspect ratio (or keyhole depth) is different from the law for fusion region depth (or melt pool depth)[3]. The depth of the fusion region is certainly larger than the keyhole depth and does not linearly scale with keyhole depth owing to the drastic difference in melt pool geometry and melt flow pattern under different laser conditions. We can thus quantify the three different melting modes of the material using the keyhole aspect ratio $e^*$ and Keyhole number Ke: conduction mode ($e^* = 0$ or Ke<1.4), transition mode ($e^* \leq 2$ or $1.4 \leq \mathrm{Ke} \leq 6.0$), and keyhole mode ($e^* > 2$ or Ke>6.0). The proposed Ke and scaling law for keyhole depth (or aspect ratio) provide a simple but effective way to identify the onset of the keyhole, which may be used in practice to reduce instability and defects arising during the AM process. We can also show the scaling behavior of the keyhole aspect ratio $e^*$:

$$e^* \propto \eta P V_s^{-\frac{1}{2}} r_0^{-\frac{3}{2}}, \qquad (3)$$

$$e^* \propto (T_1 - T_0)^{-1} \left(\pi \rho C_p\right)^{-1} \alpha^{-\frac{1}{2}}. \qquad (4)$$

Equation 3 represents a scaling law with respect to process parameters. It indicates that the keyhole aspect ratio is proportional to absorbed laser power $\eta P$ and has a power–law dependence on scan speed $V_s$ and laser spot radius $r_0$. The scaling order (exponent) is $-\frac{1}{2}$ for $V_s$ and $-\frac{3}{2}$ for $r_0$, which quantifies the effective strength of each process parameter on keyhole aspect

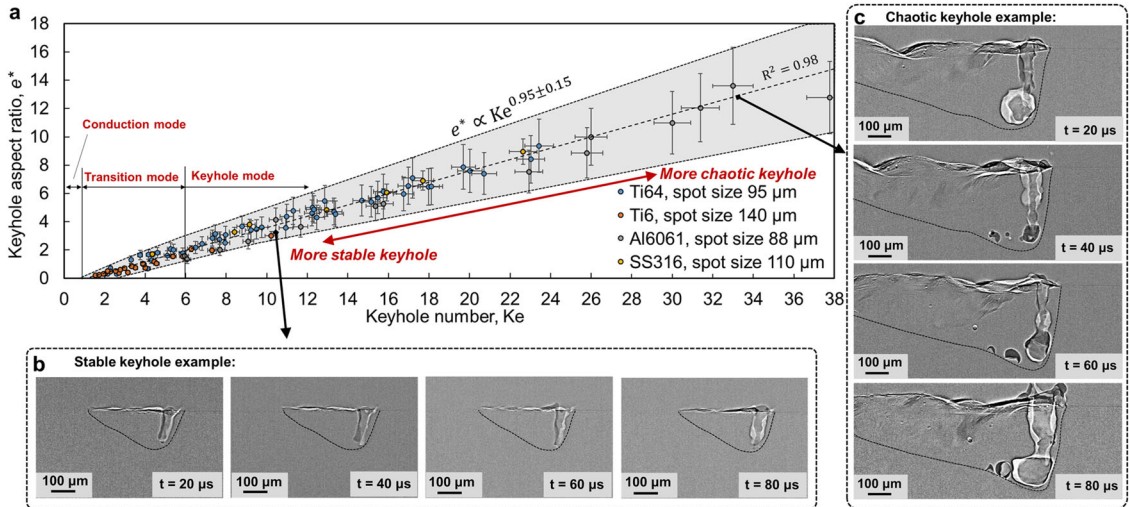

**Fig. 1 One-dimensional law for keyhole aspect ratio controlled by the Keyhole number. a** Identified scaling law, which is universal because all the data collapse to a single curve, even though they correspond to various laser power (100–520 W), scan speed (0.3–1.2 m/s), laser spot size (44–70 μm), and three different materials (Ti-6Al-4V (Ti64)[2], Aluminum alloy 6061 (Al6061), and Stainless Steel 316 (SS316)). Time-dependent keyhole depth is measured from high-speed X-ray images at a 20-μs interval for each process condition. The maximum and minimum of keyhole aspect ratio during the time period when the laser scans 2 mm length at the middle of the sample are marked as vertical error bars. Horizontal error bars indicate 3% error amount to account for uncertainties of the process parameters and material properties. **b** Operando X-ray image series with a 20-μs interval showing the keyhole and melt pool morphologies in the stable keyhole region of Al6061. Laser power is 416 W and scan speed is 0.6 m/s. **c** Operando X-ray images showing keyhole morphologies in the chaotic keyhole region of Al6061. Laser power is 520 W and scan speed is 0.3 m/s. The fusion boundary (outlined by the black dashed line) can be identified by the X-ray imaging because of the density difference inside and outside the fusion region (some operando X-ray images without dashed lines are shown in Supplementary Fig. 10).

ratio. From Eq. 3, we can also argue that the traditionally used parameters, including linear energy density $\frac{P}{V_s}$, surface energy density $\frac{P}{V_s r_0}$, and volumetric energy density $\frac{P}{V_s r_0^2}$, are not appropriate scaling parameters for the keyhole aspect ratio (Supplementary Fig. 2). Equation 4 scales the effect of material attributes, which is useful for understanding and improving the printability of metals. This scaling law explains why printing light alloys with lower liquidus temperature $T_l$ and density $\rho$ but much higher thermal diffusivity, such as aluminum or magnesium alloys, tends to create a lower keyhole aspect ratio under the same process conditions when compared to, for example, steel.

More interestingly, the variation in the keyhole aspect ratio, $\triangle e^*$, during the process represents the fluctuation and instability of the melt pool dynamics (Fig. 1 and Supplementary Movies 1–11). The melt pool instability could lead anomalies and structural defects, such as porosity, spattering, and high surface roughness. The shadowed region in Fig. 1a represents the variation in the keyhole aspect ratio $\triangle e^*$, which approximately scales with the Keyhole number as

$$\triangle e^* = 0.36 \text{Ke}^{0.86}. \tag{5}$$

Keyhole instability or variability is not observable using the traditional post-mortem characterization[3–5]. We identify a scaling law for the variation in the keyhole aspect ratio (Eq. 5) using X-ray imaging. When the Keyhole number is relatively small, e.g., Ke<16, the keyhole fluctuation is small and thus there is almost no porosity and spattering events are rare (Fig. 1b). However, when the Keyhole number is large, e.g., Ke>30, the keyhole exhibits chaotic features and becomes unstable (Fig. 1c). A small perturbation will lead to a large fluctuation of the keyhole wall and consequently pore and spatter generation. This discovery provides a clear guideline for designing the parameter space to achieve a stable keyhole.

**Energy absorption revealed by multiphysics modeling**. The absorptivity $\eta$ used in the expression of the Keyhole number is coupled with the keyhole morphology[7]. To provide insight on the energy absorption and quantify the effects of process parameters and materials, we use a high-fidelity multiphysics model that is validated with operando high-speed X-ray imaging experiments (Fig. 2). The model simulates 3D temperature, velocity, pressure, and vapor chemical species fields with 2 μm resolution and is predictive because it uses a laser ray-tracing method to calculate laser absorption and reflection within the keyhole (Supplementary Fig. 4). Furthermore, the model accounts for evaporation-induced vapor plume, which is not considered in the most multiphysics models[3,6,10] (Supplementary Method 1). The multiphase fluid mechanics is coupled with heat transfer solved on a 10-ns time scale to accurately capture laser–keyhole and keyhole–vapor interactions. The model does not consider plasma generation and laser attenuation caused by the vapor plume or plasma.

To select dominant effects and parameters for the scaling law, our model analyzes energy losses due to various mechanisms, including the reflection of the laser beam, convection caused by the vapor jet, latent heat of evaporation, surface radiation, and droplet spattering (definitions in Supplementary Method 2). The absorbed energy equals the laser energy input minus all the energy losses. Since the absorbed energy varies in a complex way depending on the keyhole morphology and temperature, our multiphysics model captures the transient keyhole morphology and the corresponding normalized power (i.e., time-dependent power divided by constant input laser power), including the absorbed part and lost parts (Fig. 3a–f). The simulation result presents four distinct regimes of this transient process: (i) melting and melt pool formation, (ii) vapor depression and keyhole growth, (iii) keyhole fluctuation, and (iv) quasi-steady state (Fig. 3a, b). These four regimes have been observed experimentally[2]. The time-resolved variations in absorbed power arise because of transient keyhole dynamics. Once the peak

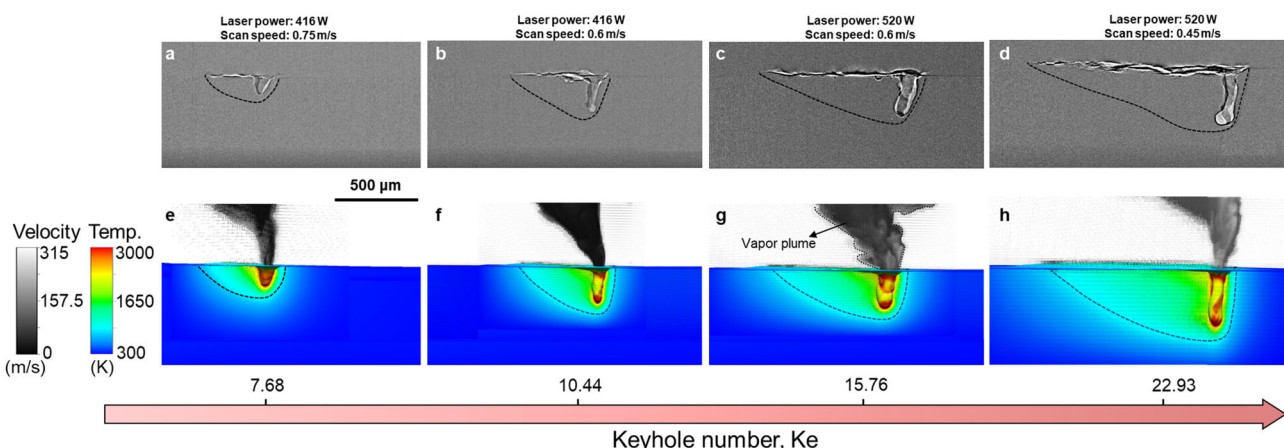

**Fig. 2 Comparison between experimental data and multiphysics simulations with various Keyhole number. a–d** Operando X-ray images of the scanning laser melting of Al6061 at steady state. **e–h** Multiphysics modeling showing temperature contour at the longitudinal cross-section and vapor velocity field above the substrate (the velocity at a spatial point is represented by an arrow and the length of the arrow is proportional to the magnitude of velocity at the specific point). The simulated high-speed vapor jet (approximately 200 m/s) impacts the keyhole interface, which is one of the sources driving the fluctuation of the keyhole and energy loss due to convection. The black dots outside the vapor plume area indicate low-speed flow (<10 m/s), specifically eddies of gas phase induced by the high-speed vapor plume.

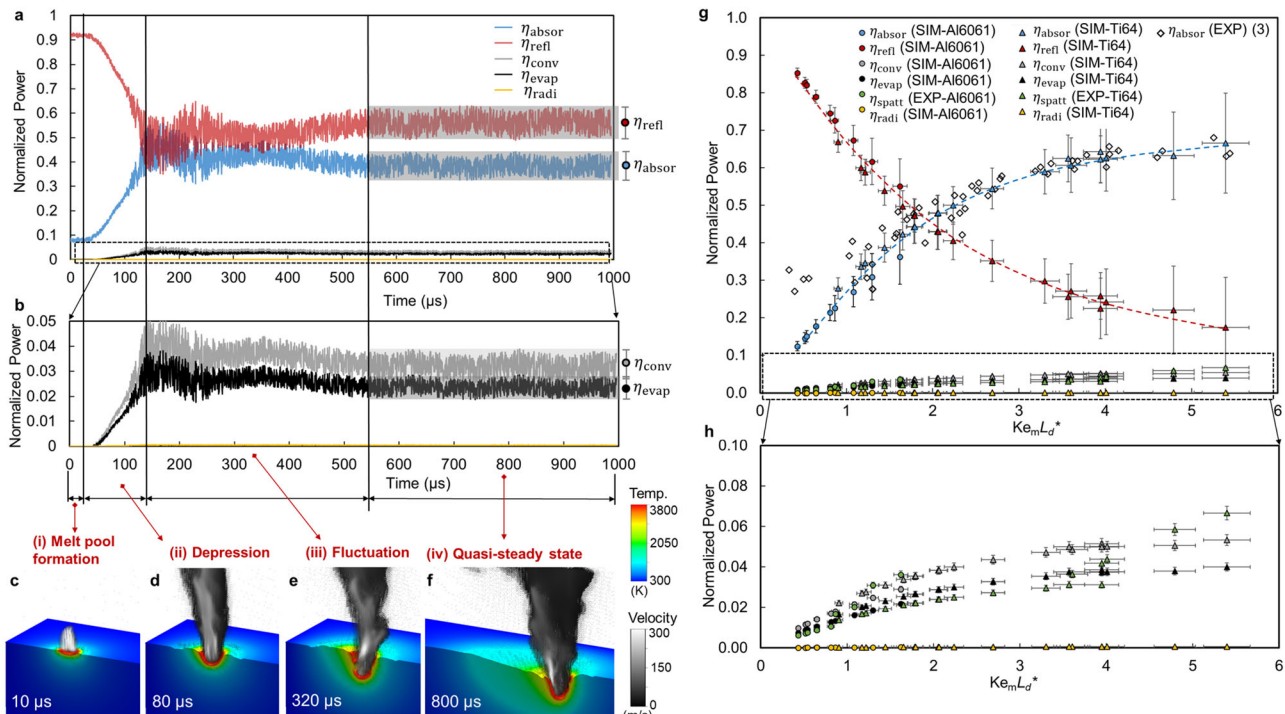

**Fig. 3 Simulation of the evolution of normalized powers. a** Transient normalized powers for a case of Ti6Al4V (laser power is 210 W and scan speed is 0.4 m/s.), including normalized absorbed power $\eta_{absor}$, reflected power $\eta_{refl}$, convected power $\eta_{conv}$, evaporated power $\eta_{evap}$, and radiated power $\eta_{radi}$. Four regimes are classified based on different features of the curves. **b** Magnification plot of transient normalized powers ranging from 0 to 0.05, to highlight the convection and evaporation terms. **c–f** Illustrative simulated keyhole morphologies and temperature fields showing the free surface and longitudinal cross-section and vapor velocity fields at different time. **g** Normalized powers with different laser powers, scan speeds, laser spot sizes, and two materials (Ti6Al4V and 6061 Aluminum alloy) (Supplementary Data 2). Experimental absorptivity data including three different materials (Ti6Al4V, Stainless steel 316, and IN718) from the literature[3] (marked as hollow diamonds) is used to validate the model prediction. The discrepancy at low normalized power is because the simulated material (Al6061) has much lower minimum absorptivity than the experimentally measured materials. **h** Magnification logarithm plot of normalized powers ranging from 0 to 0.1. The definition of the term $Ke_m L_d^*$ can be found in the main text. The vertical error bars indicate the maximum and minimum of normalized power at quasi-steady state shown in **a**, **b**. Horizontal error bars indicate 3% error amount to account for uncertainties of the process parameters and material properties.

temperature is higher than the boiling temperature of the material, the absorbed power rapidly increases as the keyhole depression deepens because of the multireflection of the laser beam between the walls of the depression. After this transition, the keyhole walls fluctuate strongly, which leads to fluctuations in all the normalized powers. The lower-frequency fluctuations on the order of $10^4$ Hz then gradually disappear, while a higher-frequency fluctuation on the order of $10^6$ Hz still exists. The magnitude of the higher-frequency fluctuation for each normalized power in the quasi-steady state is recorded as the error bars shown in Fig. 3a, b, g, h. We identify a scaling law of the process-induced absorption using 30 multiphysics simulation cases. This scaling law is validated using experimentally measured absorptivity[3] (marked on Fig. 3g). The governing parameters include the Keyhole number $Ke_m$ with minimum absorptivity $\eta_m$ of each material, i.e., absorptivity on flat melt surface[3], and a normalized diffusion length $L_d^*$ (Supplementary Method 3). The normalized diffusion length $L_d^*$ is defined as $L_d^* = \frac{\delta_z}{r_0}$, where $\delta_z$ denotes the thermal diffusion length $\delta_z = \sqrt{\frac{\alpha r_0}{V_s}}$. For $L_d^* < 1$, the thermal diffusion length is smaller than the laser beam radius, which means that the thermal energy tends to accumulate within the range of the heat source. Such a situation typically results in an elongated melt pool. For $L_d^* > 1$, the thermal diffusion length is larger than the beam radius, and the melt pool shape tends toward circular. The normalized absorbed power (absorptivity) can be expressed as

$$\eta = \eta_{absor} = 0.7\left[1 - \exp(-0.6 Ke_m L_d^*)\right], \tag{6}$$

$$Ke_m L_d^* = \frac{\eta_m P}{(T_1 - T_0)\pi\rho C_p \sqrt{\alpha V_s r_0^3}} \cdot \sqrt{\frac{\alpha}{V_s r_0}} = \frac{\eta_m P}{(T_1 - T_0)\pi\rho C_p V_s r_0^2} \cdot \tag{7}$$

The scaling parameter $Ke_m L_d^*$ represents a product of the volumetric energy density of the laser heat source and the inverse of the sensible heat of melting. The sensible heat of melting is defined as the amount of thermal energy required to change the temperature of a unit volume of a material from ambient to its melting point. The physical dimension of the sensible heat of melting is energy per volume. The validated multiphysics model can provide a detailed breakdown of the energy transfer terms, which cannot be directly measured experimentally. We conclude that the absorbed and reflected powers dominate the laser absorption since the sum of normalized powers including convection, evaporation, radiation, and spattering are around 0.1. The term reflected power describes the laser energy loss per unit time due to laser optical reflection. In keyhole mode laser melting, the reflected power is approximately proportional to the energy carried by reflected rays escaping from the keyhole (the definitions of the energy transfer terms are provided in Supplementary Method 2). Radiative energy loss is neglectable. The spattering power is measured from X-ray imaging series for bare plate experiments (Supplementary Method 2). The spattering is more frequent in laser powder bed fusion process because of the particle entrainment effect[27], which might lead to higher spattering power loss. These results rationalize that we ignore the vaporization-related parameters and mainly consider heat conduction induced by the laser energy absorption in the derivation of keyhole scaling laws (Eqs. 1 and 2). The scaling laws for the keyhole aspect ratio are derived based on thermal energy balance during the process while ignoring vaporization (Supplementary Method 3), because our simulation results reveal that the thermal energy loss due to vaporization is neglectable as compared to

absorbed and reflected energies. In addition, we hypothesize that there exists an unrevealed scaling law based on momentum conservation at the keyhole interface, which is driven by vaporization-induced recoil pressure. This particular scaling law controls keyhole morphology and its evolution.

**Inherent correlation between keyhole and pore formation.** Furthermore, we quantify inherent correlation between keyhole stability and porosity formation in metal 3D printing. We analyze the previously reported printing-induced porosity data (Fig. 4). A simple scaling law defines a two-dimensional (2D) pattern that retains descriptive capabilities for porosity data collected from six independent studies[6,21,28–31] spanning ten process parameter sets and material properties. This scaling law is expressed as

$$\phi = 0.10 \,\mathrm{erfc}\,(NED - 1.51) + 0.01 \,\mathrm{erf}\left(Ke_m L_d^* - 2.32\right) + 0.055, \tag{8}$$

where $\phi$ describes the volume fraction of the porosity. A Gaussian error function, i.e., $\mathrm{erf}(x) = \frac{1}{\sqrt{\pi}}\int_{-x}^{x}\exp(-t^2)\mathrm{d}t$, and its complementary function, defined as $\mathrm{erfc}(x) = 1 - \mathrm{erf}(x)$, completely describes the data, which implies that the origin of porosity formation might be related to specific Gaussian random processes. Two dimensionless numbers dominate the low-dimensional pattern with good accuracy (coefficient of determination, $R^2$, is 0.92). The first dimensionless number is the well-known normalized energy density (NED)[4–6], defined by

$$NED = \left(\frac{\eta_m P}{V_s HL}\right)\left(\frac{1}{\rho C_p(T_1 - T_0)}\right), \tag{9}$$

where $H$ is the hatch spacing and $L$ is the powder bed thickness in laser powder bed fusion process. The normalized energy density represents the ratio of effective energy density within the powder bed to sensible heat of melting. Interestingly, the product of the Keyhole number $Ke_m$ (with minimum absorptivity $\eta_m$) and the normalized diffusion length $L_d^*$ appears again as the second independent parameter, despite the fact that this porosity model is completely independent from the previous case of keyhole dynamics. Although powder effects affect the laser energy absorptivity, especially in the low-power range[10], the Keyhole number does not include the layer thickness of the powder bed $L$ because our aim is to generate a simplified relationship, and the existence of a powder bed has only a small effect on the keyhole. This is consistent with experimental observations[2]. The appearance of the Keyhole number in these seemingly unrelated problems (keyhole morphology and porosity) implies that the porosity generation is analogous to keyhole formation and will not appear below a critical Keyhole number.

The first term of the scaling law for porosity is related only to NED. This term fits data with low NED, which is a regime characterized by lack-of-fusion defects (Fig. 4a). When NED exceeds the critical value (approximately 5), lack-of-fusion porosity can be avoided and the efficacy of NED as a predictor of porosity decreases. Meanwhile, the second term and the constant part of the scaling law begin to represent the observed data well (Fig. 4b). Since the Keyhole number $Ke_m$ and normalized diffusion length $L_d^*$ are determinant parameters here, this part of the scaling law can be understood as higher input energies related to sensible heat of melting resulting in keyhole porosity, and the scaling law quantifies the effect of parameters. The combination of the two mechanisms as a 2D scaling law is visualized (shown as a mesh surface) along with experimental data points (colored dots) in Fig. 4c. The proposed scaling law (Eq. 8 and Fig. 4c) is advantageous over the traditional NED

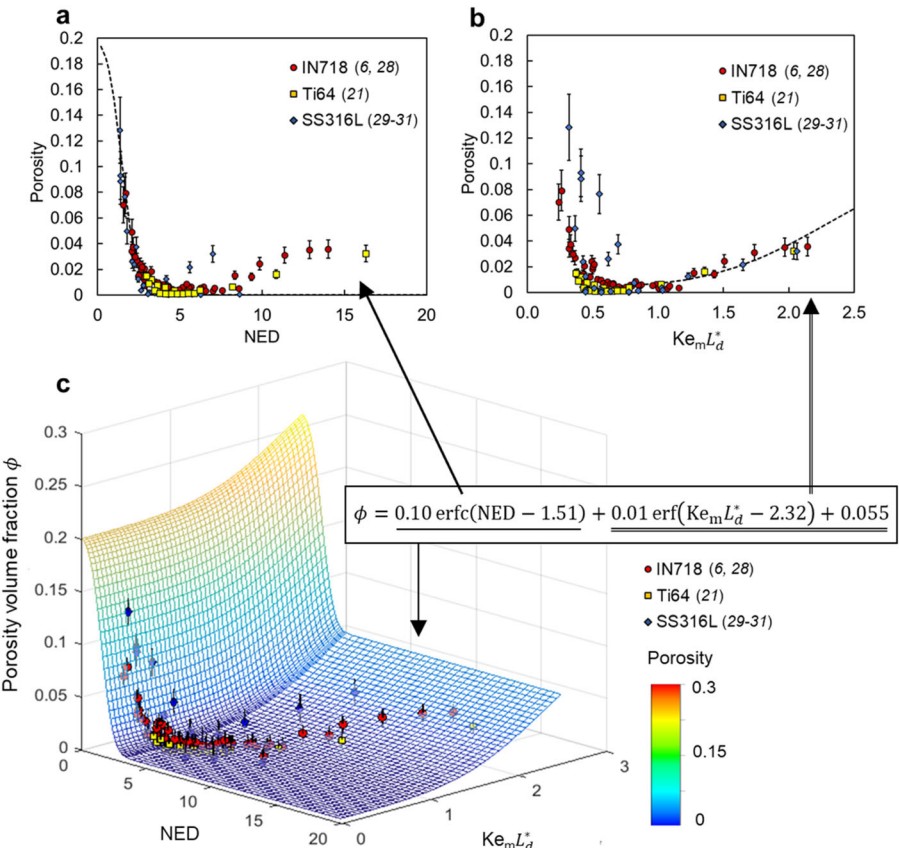

**Fig. 4 Two-dimensional (2D) scaling law for volume fraction of porosity. a** The first term of the scaling law associated with the normalized energy density (NED), which quantifies porosity formed via the lack of fusion process. **b** The second and the constant terms of the scaling law associated with the product of Keyhole number, $Ke_m$, and normalized diffusion length, $L_d^*$, which quantifies the keyhole porosity. **c** 2D scaling law combining both lack of fusion and keyhole porosity. Methods of measuring the porosity and computing error bars are provided in the literature[6,21,28–31]. This reduced function space is much more easily visualized and interpreted than the original high-dimensional problem in which the porosity is controlled by ten parameters. By making it easy to visualize, we enable more intuitive optimization and use. The error bars account for standard errors of experimental measurements.

description (Eq. 9 and Fig. 4a) because of its universality. Equation 8 captures minimal porosity (near full density) conditions for three different AM materials and is able to be extended for more materials and therefore provides a simple transition rule that translates optimal process parameters from one material (or existing materials) to another (or new materials). However, the traditional NED description (Fig. 4a) captures differences in near-full density conditions for different materials but fails to provide any relation between them. Thus, it is not helpful to approximate the near full density condition for a new material based on the existing data.

## Discussion

Although the proposed scaling laws are verified in laser-based AM processes, the methodology and results can be applied to the broader field of energy beam research with some necessary modifications. Possible areas of applicability include laser welding and cladding[32], abrasive waterjet milling[33], and arc/plasma/electron beam-based manufacturing[34]. For example, some modifications would be required to apply these laws to electron beam-based AM processes. We hypothesize two reasons: first, the energy absorption mechanism of electron–material interaction (i.e., electron collision) is different from laser–material interaction (i.e., laser multiple reflection), which could lead to a different form of energy absorptivity in the scaling law. Second, electron beam fusion is typically conducted in a vacuum environment

instead of an Argon/Helium shielding environment used in most laser-based AM processes. The vacuum environment alters the vapor plume dynamics and resulting keyhole size and morphology.

Although the X-ray experiments are conducted on a bare plate in this work, a previous study reports that the keyhole depth does not change significantly by adding a powder layer between 50 and 100 μm deep[2]. Recently, another study[18] reconfirmed this statement by concluding that the boundary of the keyhole porosity regime in process parametric space varies only slightly between the bare plate and powder bed conditions. Thus, we believe that the proposed universal scaling is valid for powder bed AM process although some fitting constants might need to be adjusted to account for uncertainties and biases caused by the powder layer[35]. In this study, we apply the scaling laws to a broad set of data and commercial powder bed AM machines. The results demonstrate that there is a significant correlation between the proposed Keyhole number and porosity formed during powder bed multitrack and multilayer AM processes. These results demonstrate the effectiveness of the proposed scaling laws in practice for AM processes, which might have potential to be used for quantifying other keyhole-related defects, such as spattering and soot.

The keyhole scaling laws (Eqs. 1, 2, and 5) are valid for a quasi-steady-state melt pool, meaning that the melt pool is well developed, and the melt pool size and temperature distribution are approximately unchanging in time (although some fluctuations exist in practice due to the highly dynamic multiphase flow).

Several experimental and simulation results have shown that in powder bed AM processes the melt pool can reach the quasi-steady state within a few milliseconds (or after laser scanning 1–2 mm from the start)[36,37]. Therefore, even though AM in practice involves multitrack and multilayer build conditions, most of the solidified region is created while the melt pool is in a quasi-steady-state condition for which the proposed scaling laws are valid to control the keyhole size and stability. However, it is worth noting that the proposed keyhole scaling law is inapplicable to a transient melt pool appearing at the starting and end positions of the scanning track or laser turning locations, where the keyhole exhibits different size and morphology as it in quasi-steady state[38].

This study provides experimental data obtained using ultrahigh-speed synchrotron X-ray imaging and simulation data created with a well-tested multiphysics model. These data are particularly useful for the development and validation of statistical and machine learning models. Multiple scaling laws are derived to quantify keyhole aspect ratio and keyhole instability, which is not observable using the traditional post-mortem characterization. A developed multiphysics model elucidates energy losses due to various physical mechanisms. We discover an inherent quantitative relation between keyhole instability and porosity formation, which provides better predictive capability than classical qualitative descriptions. The keyhole dynamics embedded by these compact scaling laws enables to quantify process-induced phenomena in metal 3D printing. These concise laws are able to reduce complex, highly multivariate problem spaces into descriptions involving just a few physically interpretable parameters. The dominant laws for keyhole stability and porosity reveal dynamics underlying the laser–metal interaction processes, which can potentially lead to quantitative predictive models for controlling defect generation in metal 3D printing. The reduction of high-dimensional parameter space means that fewer experiments will be required to determine optimal processing conditions for new materials and thus ease the Edisonian burden endemic among current metal 3D printing practitioners.

## Methods

**Materials**. We manufactured 50 mm-by-3 mm-by-0.75 mm samples of aluminum (Al6061, McMaster-Carr, USA) and 50 mm-by-3 mm-by-0.5 mm samples of stainless steel (SS316, McMaster-Carr, USA) from as-received plates using conventional manufacturing methods (cutting and milling). The samples were polished on all sides before being loaded into the vacuum chamber at the beamline.

**Selective laser melting apparatus**. We built a custom selective laser melting apparatus by integrating an ytterbium fiber laser (YLR-500-AC, IPG, USA), a galvo laser scanner (IntelliSCANde30, SCANLAB GmbH., Germany), a vacuum chamber, and multiple translational motor stages. Ar gas is filled into the chamber to prevent the potential oxidation of the metals. The laser wavelength was 1070 nm and the maximum power was 540 W. The laser source was operated in single mode, providing a Gaussian beam profile. At the focal plane, the laser spot size was ≈50 μm. In this study, samples were positioned at a certain distance below the focal plane to achieve larger laser spot sizes on sample surface (1.5 mm for a spot size of 84 μm, 1.8 mm for 93 μm, 2.5 mm for 114 μm, and 3.5 mm for 144 μm). Uncertainty of the laser spot size on sample is about ±5 μm. Single track laser melting experiments were performed on the samples under various laser powers (208–520 W) and scan speeds (0.3–1.2 m/s).

**High-speed X-ray imaging**. The high-speed X-ray imaging experiments were performed at beamline 32-ID-B of the Advanced Photon Source at Argonne National Laboratory. A short period (18 mm) undulator with the gap set to 12 mm was used to generate polychromatic X-rays with the first harmonic energy centered at 24.7 keV. The X-rays were allowed to pass through the sample while the laser was traversing across the top surface. The propagated X-ray signal was converted to visible light using a LuAG:Ce scintillator (100-μm thickness) and recorded with a high-speed camera (Photron FastCam SA-Z, USA) after passing a 45° reflection mirror, a relay lens, and a ×10 objective lens. The nominal spatial resolution of the imaging system was 1.93 μm/pixel. We recorded high-speed X-ray images at frame rates between 20,000 and 50,000 frames per second with exposure times between 1

and 40 μs, with higher exposure times used for stainless steel samples. A series of delay generators were used to trigger the X-ray shutters, laser system, and high-speed camera sequentially. More details of the high-speed X-ray imaging experiments on laser AM are provided elsewhere[12,14].

**Data processing and quantification**. The images of keyholes and melt pools were processed and analyzed using ImageJ[39]. Each image stack includes a time series of images. For each, we first duplicated the original image stack (A) to create an identical stack (B). Second, we duplicated the first slice of stack A and the last slice of the stack B. Third, we divided image stack A pixel-wise and slice-wise for each slice by stack B to reduce background noise and increase contrast of the interesting structure features. Fourth, we used a despeckling median filter to further eliminate speckle noise. Finally, we adjusted the brightness and contrast manually to enhance contrast and ease visualization and dimension measurement. We quantified the size of the melt pool and keyhole based on contours of attenuation contrast. This involved manual inspection and measurement of each image in all the image stacks in ImageJ, recording the XY coordinates describing length and depth of the melt pool and keyhole for each image. The maximum, minimum, mean, and standard deviation for each stack were calculated.

Wavelet transform (WT)[40] was used to analyze the normalized power curves and quantify the frequency as it changes over time. The order of magnitude of the dominant frequencies was approximated from the WT analysis. The WT was implemented in MATLAB 2019 (MathWorks, USA). The time–power curves were converted to time–frequency spectrums. The mother wavelets were the generalized Morse wavelets[41]. The sampling frequency used was 1788 kHz.

## Data availability

All data are available in the main text or the Supplementary Information. The data consist of X-ray imaging movies and Excel files, including measured keyhole dimensions and simulated transient powers.

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

## Acknowledgements

We gratefully acknowledge the computing resources provided on Bebop, a high-performance computing cluster operated by the Laboratory Computing Resource Center at Argonne National Laboratory. We also thank the Center for Hierarchical Materials Design (CHiMaD), in particular, our ongoing work with Lyle E. Levine has been synergistic. W.K.L., Z.G., O.L.K., and L.F. were supported by the National Science Foundation (NSF) through grants CMMI-1762035 and CMMI-1934367. O.L.K. acknowledges support through the NSF Graduate Research Fellowship under Grant No. DGE-1324585. N.P., C.Z., and T.S. would like to acknowledge Laboratory Directed Research and Development (LDRD) funding from Argonne National Laboratory, provided by the Director, Office of Science, of the U.S. Department of Energy under Contract No. DE-AC02-06CH11357; the work by O.H. was performed under financial assistance award 70NANB14H012 from U.S. Department of Commerce, National Institute of Standards and Technology as part of the CHiMaD. This research used resources of the Advanced Photon Source, a U.S. Department of Energy (DOE) Office of Science User Facility operated for the DOE Office of Science by Argonne National Laboratory under Contract No. DE-AC02-06CH11357.

## Author contributions

O.H., W.K.L., and Z. G. supervised the project. Z.G., O.L.K., and O.H. analyzed the results and wrote the first manuscript. N.P., C.Z., and T.S. designed and conducted the experiments. Z.G. implemented the multiphysics modeling and dimensional analysis. L.F. conducted post-processing of X-ray images.

## Competing interests

The authors declare no competing interests.
