## [Peer Review File · Nature Communications]

REVIEWER COMMENTS

Reviewer #1 (Remarks to the Author):

This paper explores the formation of key holes in scenarios which approximate current class additive manufacturing. Our understanding of the keyhole formation is somewhat underplayed in the introduction to this work. In fact others have explored laser-material interaction in similar setups using high energy X-Ray techniques. For example Richter et al. (uncited in this work) explored phenomena through similar apparatus as did Leung et al. (who are cited in this work).

This work diverges (and is novel) in that it explores key hole scaling arrangements. The formation and collapse of keyholes are well understood and vapour pressure builds up within a recess in the process vector which maintains integrity (stays open) as the process continues. Understanding key hole formation is important as it contributes to the stability of energy beam processes in AM (and welding techniques).

On this point;

1. The authors investigate laser only. Would we expect a similar response through electron beam techniques?
- 2, Other authors used powder beds in their experiments and as such were able to better simulate 'real' process phenomena. It is questionable whether the scaling laws identified here will translate to higher stability in the 'noisier' circumstance associated with additive manufacturing. Is relating this to 3D printing a bit rich with the current results?

The study is undertaken systematically and voluminous data is provided which support observations across three materials systems. Credible trends are observed which would be useful to other researchers in our field hence the mood of this review is overall supportive.

In framing the work though the authors may wish to target the broader energy beam research network. Svenungsson's review (while less useful in itself) may provide stimulating literature in other areas which can add to the utility of the present work. In addition good commentary can be found in Axinte and Billingham's work on the footprint of energy beam processes.

Quite clearly the 'minimum' energy required to form a key hole of an appropriate depth is always the target. Over provision of energy runs the risk of instability and defects (much work has been done in this space). AS such can the authors dedicate some discussion to the practical use of this know-how. It should be noted that the present work takes place in a vacuum and on a solid metal surface. It does not represent the 3D printing scenario well.

The authors may also wish to consider ancillary factors (substrate temperature, environmental pressure) which are also known to effect the formation and stability of keyholes. Of course fixed energies per unit length are one way of considering the process-material relationships but also emergent scanning strategies may challenge this approach.

I am supportive of this work progressing for further consideration as I am convinced there is a valuable contribution however there is significant work to be undertaken to ensure it will reach the correct readership and make clear the novelty over and above 'classical' contributions made by Stein amongst others.

Reviewer #2 (Remarks to the Author):

The proposed scaling laws for keyhole stability and porosity are indeed simple and useful. The authors have done a great job in introducing a new dimension number - the Keyhole number, K_e , to identify different processing welding modes for three types of materials as the experiments are based on bare plates. Though they have applied the scaling laws using other data and powder bed machines to prove their application, and the results look very promising. I think this work is very nicely executed, however, there are several aspects of the manuscript requiring further clarifications, see my comments below.

Given that additive manufacturing involves multi-hatch and multi-layer build, your work is based on a few mm single layer weld track. Do these laws apply to large scale printing?

Can the authors comment on the limitations of such laws? In a real AM machine, the effective layer thickness in AM machines is usually $> 50 \mu\text{m}$ as quantified by Mahmoodkhani et al. *Prog Addit Manuf* 4, 109–116 (2019). <https://doi.org/10.1007/s40964-018-0064-0>.

p.3 and p.4: Can you double-check the exponent of r_0 in equation (1), equation (3), and p.4 line 1?

p.3 Figure 1 How did you determine the liquid/solid interface – it is unclear to me based on the contrast of the operando X-ray images. Can you show those images without the dotted line?

p.4 line 35: Based on the supplementary text and equations, the authors said the simulation model is a multi-physics model but line 35 stated that is a '...multiphase model...'. Can you verify the claim?

p.5 Figure 2A-D: Same comment as Figure 1.

p.5 Figure 2E-H: Can you also explain why there are black dots outside the vapour plume area?

p.5 lines 22 - 25: 'The magnitude of the higher-frequency fluctuation...', it's not clear how you quantify the frequency fluctuation. This should be included in the method section. Can you explain this further?

p.5 line 30: Give that the normalized diffusion length is an important parameter to this study, I would recommend that the authors should discuss it in the main text rather than in the supplementary file.

p.6 line 13: Can you clarify the term 'sensible heat melting' as this is not a common term in the field?

Further, can you comment on how reflected powers involved in the laser absorption mechanism? 'Reflected powers' is not a common term in the field and it was neither discussed in detail within the main text nor the supplementary information.

p.6 line 22: 'This rationalizes the fact that the vaporization-related parameters, such as boiling point and latent heat of vaporization, are not necessarily included in the scaling laws.' This is an interesting statement given that the simulation model you used requires the vapourisation parameter. Without the vapourisation parameter, it would have not been possible to confirm that the reflected powers are involving in the laser absorption mechanism. Are the authors ruling out the need for considering laser-vapor plume interaction when it comes to the prediction of the keyhole morphology and defect generation?

p.8, line 10: ‘...existence of a powder bed has only a small effect on the keyhole...’ I think the authors should confirm whether these scaling laws would apply thick powder bed 50 – 150 μm otherwise, this may mislead readers.

In the supplementary information, it’s unclear to me how you quantify the length of the melt pool. Can you elaborate on that?

Regards

Chu Lun Alex Leung

Reviewer #3 (Remarks to the Author):

Dear Authors,

I believe that what has been achieved here is nothing short of outstanding. The ability to explain how and why keyholing progresses via high-speed synchrotron X-ray imaging is a key outcome of the work. I am skeptical, or rather not as thrilled with the quantification. But the really fun aspect of this paper was reading about the manner in which keyhole formation occurs. Nothing to add or subtract from this paper. I would just suggest to make the diagrams a little "big and barky". Otherwise, nothing to nitpick or complain about.

Reviewer 1

1) This paper explores the formation of key holes in scenarios which approximate current class additive manufacturing. Our understanding of the keyhole formation is somewhat underplayed in the introduction to this work. In fact, others have explored laser-material interaction in similar setups using high energy X-Ray techniques. For example, Richter et al. (uncited in this work) explored phenomena through similar apparatus as did Leung et al. (who are cited in this work).

*We thank the reviewer for their time and effort in reviewing this manuscript and pointing out the important work uncited in the manuscript. We have carefully gone through more related articles and added them (including Richter et al.) in the **Introduction** of the revised manuscript:*

“These high-energy x-ray imaging experiments have been conducted in laser melting of bare plate (12), powder bed (13-18), and powder flow (19).”

The four references added are:

- 12. M. Miyagi, J. Wang, Keyhole dynamics and morphology visualized by in-situ X-ray imaging in laser melting of austenitic stainless steel. J. Mater. Process. Technol. 116673 (2020).*
- 17. B. Richter, N. Blanke, C. Werner, N. D. Parab, T. Sun, F. Vollertsen, F. E. Pfefferkorn, High-speed X-ray investigation of melt dynamics during continuous-wave laser remelting of selective laser melted Co-Cr alloy. CIRP Annals. 68, 229-232 (2019).*
- 18. C. Zhao, N. D. Parab, X. Li, K. Fezzaa, W. Tan, A. D. Rollett, T. Sun, Critical instability at moving keyhole tip generates porosity in laser melting. Science. 370, 1080-1086 (2020).*
- 19. S. J. Wolff, S. Webster, N. D. Parab, B. Aronson, B. Gould, A. Greco, T. Sun, In-situ observations of directed energy deposition additive manufacturing using high-speed x-ray imaging. JOM, 1-12 (2020).*

2) This work diverges (and is novel) in that it explores keyhole scaling arrangements. The formation and collapse of keyholes are well understood and vapour pressure builds up within a recess in the process vector which maintains integrity (stays open) as the process continues. Understanding keyhole formation is important as it contributes to the stability of energy beam processes in AM (and welding techniques).

On this point;

1. The authors investigate laser only. Would we expect a similar response through electron beam techniques?

That is a very interesting point. We believe that some modifications would be required to apply these laws to electron beam-based AM processes. We hypothesize two reasons here: first, the energy absorption mechanism of electron-material interaction (i.e., electron collision) is different from laser-material interaction (i.e., laser multiple reflection), which could lead to a different form of energy absorptivity in the scaling law. Second, the electron beam fusion is typically conducted in a vacuum environment instead of an Argon/Helium shielding environment used in most laser-based AM processes. The vacuum environment significantly affects vapor plume dynamics and resulting keyhole size and morphology.

*We added a short discussion in the **Discussion** of the revised manuscript:*

“For example, some modifications would be required to apply these laws to electron beam-based AM processes. We hypothesize two reasons: first, the energy absorption mechanism of electron-material interaction (i.e., electron collision) is different from laser-material interaction (i.e., laser multiple reflection), which could lead to a different form of energy absorptivity in the scaling law. Second, electron beam fusion is typically conducted in a vacuum environment instead of an Argon/Helium shielding environment used in most laser-based AM processes. The vacuum environment alters the vapor plume dynamics and resulting keyhole size and morphology.”

2. Other authors used powder beds in their experiments and as such were able to better simulate 'real' process phenomena. It is questionable whether the scaling laws identified here will translate to higher stability in the 'noisier' circumstance associated with additive manufacturing. Is relating this to 3D printing a bit rich with the current results?

We agree that this is an important point, and it should be further clarified. We mentioned in the caption of Figure 1 of the manuscript “Although the x-ray experiments are conducted on a bare plate, a previous study reports that the keyhole depth does not change significantly by adding a thin layer of powder (R. Cunningham et al., Science 363, 849-852 (2019).).” Recently, another published work (C. Zhao et al., Science. 370, 1080-1086 (2020).) reconfirms our statement. Zhao et al. concluded that “We found that the boundary of the keyhole porosity regime in power-velocity space is sharp and smooth, varying only slightly between the bare plate and powder bed (emphasis ours)”. Thus, we believe that the proposed universal scaling is valid for powder bed AM processes, although some fitting constants might need to be adjusted to account for uncertainties and biases caused by the powder layer. This will be an incremental future work.

*Furthermore, we have applied the scaling laws using other data and powder bed AM machines in the subsection: **Inherent correlation between keyhole and pore formation** of the revised manuscript. The results demonstrate that there is a significant correlation between the proposed Keyhole number and porosity formed during powder bed multitrack and multilayer AM processes. We believe these results show the effectiveness of the proposed scaling laws in practice for AM processes, and have the potential to be used for quantifying other keyhole-related defects such as spattering and soot.*

*We added a short discussion in the **Discussion** of the revised manuscript:*

“Although the x-ray experiments are conducted on a bare plate in this work, a previous study reports that the keyhole depth does not change significantly by adding a powder layer between 50 μm and 100 μm deep (2). Recently, another study (18) reconfirms this statement by concluding that the boundary of the keyhole porosity regime in process parametric space varies only slightly between the bare plate and powder bed conditions. Thus, we believe that the proposed universal scaling is valid for powder bed AM process although some fitting constants might need to be adjusted to account for uncertainties and biases caused by the powder layer (35). In this study, we apply the scaling laws to a broad set of data and commercial powder bed AM machines. The results demonstrate that there is a significant correlation between the proposed Keyhole number and porosity formed during powder bed multitrack and multilayer AM processes. These results demonstrate the effectiveness of the proposed scaling laws in practice for AM processes, which might have potential to be used for quantifying other keyhole-related defects such as spattering and soot.”

The three references are:

2. R. Cunningham, C. Zhao, N. Parab, C. Kantzos, J. Pauza, K. Fezzaa, T. Sun, A. D. Rollett, Keyhole threshold and morphology in laser melting revealed by ultrahigh-speed x-ray imaging. *Science* 363, 849-852 (2019).

18. C. Zhao, N. D. Parab, X. Li, K. Fezzaa, W. Tan, A. D. Rollett, T. Sun, Critical instability at moving keyhole tip generates porosity in laser melting. *Science*. 370, 1080-1086 (2020).

35. Y. Mahmoodkhani, U. Ali, S. I. Shahabad, A. R. Kasinathan, R. Esmaeilizadeh, A. Keshavarzkermani, E. Toyserkani, On the measurement of effective powder layer thickness in laser powder-bed fusion additive manufacturing of metals. *Prog. Addit. Manuf.* 4, 109-116 (2019).

3) The study is undertaken systematically and voluminous data is provided which support observations across three materials systems. Credible trends are observed which would be useful to other researchers in our field hence the mood of this review is overall supportive.

We thank the reviewer for the encouraging comment. We agree that the provided data and movies will be very useful to researchers in the field. In our opinion, these data are particularly useful for the validation of simulation models and machine learning models.

4) In framing the work though the authors may wish to target the broader energy beam research network. Svenungsson's review (while less useful in itself) may provide stimulating literature in other areas which can add to the utility of the present work. In addition good commentary can be found in Axinte and Billingham's work on the footprint of energy beam processes.

*We added a short discussion in the **Discussion** of the revised manuscript:*

“Although the proposed scaling laws are verified for laser-based AM processes, the methodology and results can be applied to the broader field of energy beam research with some necessary modifications. Possible areas of applicability include laser welding and cladding (32), abrasive waterjet milling (33), and arc/plasma/electron beam-based manufacturing (34).”

The three new references are:

32. J. Svenungsson, I. Choquet, A. F. Kaplan, Laser welding process—a review of keyhole welding modelling. *Phys. Procedia*. 78, 182-191 (2015).

33. M. C. Kong, S. Anwar, J. Billingham, D. A. Axinte, Mathematical modelling of abrasive waterjet footprints for arbitrarily moving jets: part I—single straight paths. *Int. J. Mach. Tools Manuf.* 53, 58-68 (2012).

34. J. Norrish, *Advanced welding processes*. (Elsevier, 2006).

5) Quite clearly the 'minimum' energy required to form a key hole of an appropriate depth is always the target. Over provision of energy runs the risk of instability and defects (much work has been done in this space). AS such can the authors dedicate some discussion to the practical use of this know-how. It should be noted that the present work takes place in a vacuum and on a solid metal surface. It does not represent the 3D printing scenario well.

We agree that the ‘minimum’ energy required to form a keyhole of an appropriate depth is a target. The proposed Keyhole number (Ke) and scaling law for keyhole depth (or aspect ratio) provide a very simple but effective way to identify the onset of the keyhole. Specifically, the melting mode is conduction (there is no keyhole) if the Ke is less than 1.4. The melting mode is transition, in which the keyhole is swallow and typically stable, if the Ke is greater than 1.4 and less than 6. Since the proposed keyhole scaling law is valid across different process conditions and three materials, it can be applied to many practical circumstances.

We clarify that the present work takes place in an Ar gas filled chamber, the same as many practical/commercial powder bed AM applications. The effect of the powder bed has been discussed in the previous response.

*A clarification has been added in the second paragraph of the **Results** of the revised manuscript:*

“The proposed Ke and scaling law for keyhole depth (or aspect ratio) provide a simple but effective way to identify the onset of the keyhole, which may be used in practice to reduce instability and defects arising during the AM process.”

6) The authors may also wish to consider ancillary factors (substrate temperature, environmental pressure) which are also known to effect the formation and stability of keyholes. Of course fixed energies per unit length are one way of considering the process-material relationships but also emergent scanning strategies may challenge this approach.

In this work, those ancillary factors (substrate temperature, environmental pressure) are fixed in all the experiments. Thus, their effects cannot be quantified. We could consider more ancillary factors in the future.

*A clarification has been added in the first paragraph of the **Results** of the revised manuscript:*

“Some ancillary factors, such as substrate temperature and environmental pressure, are fixed in all the experiments. Their effects are thus not quantified in this work.”

7) I am supportive of this work progressing for further consideration as I am convinced there is a valuable contribution however there is significant work to be undertaken to ensure it will reach the correct readership and make clear the novelty over and above 'classical' contributions made by Stein amongst others.

We thank the reviewer’s positive and constructive comments. This work provides new experimental data using ultrahigh-speed synchrotron x-ray imaging and new simulation data using a well-tested multiphysics model. These data are particularly useful for the validation of statistical and machine learning models. Multiple scaling laws are discovered to quantify keyhole aspect ratio and instability for the first time. We emphasize that the keyhole instability is not observable using the traditional post-mortem characterization. A newly developed multiphysics model elucidates energy losses due to various physical mechanisms. In addition, we discover an inherent quantitative relation between keyhole instability and porosity formation, which provides unprecedented predictive capability as compared to classical qualitative descriptions.

*A short discussion about the novelty of this work has been added in the **Discussion** of the revised manuscript:*

“This study provides new experimental data obtained using ultrahigh-speed synchrotron x-ray imaging and new simulation data created with a well-tested multiphysics model. These data are particularly useful for the development and validation of statistical and machine learning models. Multiple scaling laws are derived to quantify keyhole aspect ratio and keyhole instability, which is not observable using the traditional post-mortem characterization. A newly developed multiphysics model elucidates energy losses due to various physical mechanisms. We discover an inherent quantitative relation between keyhole instability and porosity formation, which provides better predictive capability than classical qualitative descriptions.”

Reviewer 2

1) The proposed scaling laws for keyhole stability and porosity are indeed simple and useful. The authors have done a great job in introducing a new dimension number - the Keyhole number, K_e , to identify different processing welding modes for three types of materials as the experiments are based on bare plates. Though they have applied the scaling laws using other data and powder bed machines to prove their application, and the results look very promising. I think this work is very nicely executed, however, there are several aspects of the manuscript requiring further clarifications, see my comments below.

We appreciate the reviewer's time and effort in reviewing this manuscript. We thank the reviewer for their positive comments.

2) Given that additive manufacturing involves multi-hatch and multi-layer build, your work is based on a few mm single layer weld track. Do these laws apply to large scale printing?

In this study, x-ray imaging is conducted at a 2 mm length window in the middle of each sample (the sample is 50 mm long in the scan direction). Thus, the keyhole scaling laws (Equations 1,2 and 5) are determined for, and valid for, quasi-steady state melt pools, meaning that the melt pool is well-developed and the melt pool size and temperature distribution are approximately unchanging in time (although some fluctuations exist in practice due to variability in the AM process). Several experimental and simulation results in the literature have shown that in powder bed AM processes the melt pool can achieve the quasi-steady state within a few milliseconds (or after scanning 1-2 millimeters from the start) (see, e.g., J. C. Heigel et al. (2018) and T. Mukherjee et al. (2018)). Therefore, even though additive manufacturing in practice involves multi-hatch and multi-layer build conditions, most of the solidified region is created while the melt pool is in a quasi-steady state condition, for which the proposed scaling laws are valid to control the keyhole size and stability. It is worth noting that our keyhole scaling law is inapplicable to a transient melt pool appearing at the very beginning of the process or when the laser beam changes directions suddenly. In these situations, the keyhole depth might suddenly increase and then gradually decrease until the melt pool achieves a quasi-steady state (Martin et al. (2019)).

For the porosity scaling problem, we considered key parameters such as hatch spacing and powder bed thickness in Equations 8 and 9 so the scaling law can be applied to multitrack and multilayer AM process.

*We added a short discussion in the **Discussion** of the revised manuscript:*

“The keyhole scaling laws (Equations 1,2 and 5) are valid for a quasi-steady state melt pool, meaning that the melt pool is well-developed and the melt pool size and temperature distribution are approximately unchanging in time (although some fluctuations exist in practice due to the highly dynamic multiphase flow). Several experimental and simulation results have shown that in powder bed AM processes the melt pool can reach the quasi-steady state within a few milliseconds (or after laser scanning 1-2 millimeters from the start) (36, 37). Therefore, even though AM in practice involves multitrack and multilayer build conditions, most of the solidified region is created while the melt pool is in a quasi-steady state condition for which the proposed scaling laws are valid to control the keyhole size and stability. However, it is worth noting that the proposed keyhole scaling law is inapplicable to a transient melt pool appearing at the starting and end positions of the scanning track or laser turning locations, where the keyhole exhibits different size and morphology as it in quasi-steady state (38).”

The three new references added are:

36. J. C. Heigel, B. M. Lane, Measurement of the melt pool length during single scan tracks in a commercial laser powder bed fusion process. *J. Manuf. Sci. Eng.* 140 (2018).
37. T. Mukherjee, T. DebRoy, Mitigation of lack of fusion defects in powder bed fusion additive manufacturing. *J. Manuf. Process.* 36, 442-449 (2018).
38. A. A. Martin, N. P. Calta, S. A. Khairallah, J. Wang, P. J. Depond, A. Y. Fong, C. J. Tassone, Dynamics of pore formation during laser powder bed fusion additive manufacturing. *Nat. Commun.* 10, 1-10 (2019).

*3) Can the authors comment on the limitations of such laws? In a real AM machine, the effective layer thickness in AM machines is usually $> 50 \mu\text{m}$ as quantified by Mahmoodkhani et al. *Prog Addit Manuf* 4, 109–116 (2019). <https://doi.org/10.1007/s40964-018-0064-0>.*

One limitation of the proposed scaling laws is that they are only valid for quasi-steady state melt pool: inapplicable to a transient melt pool appearing briefly at the beginning of the process or when the laser beam changes direction. This clarification has been added in the revised manuscript.

*Although the x-ray experiments are conducted on a bare plate, a previous study reports that the keyhole depth does not change significantly by adding a thin layer of powder (R. Cunningham et al., *Science* 363, 849-852 (2019)). Recently, another study was published (C. Zhao et al., *Science*. 370, 1080-1086 (2020)) concluding that “We found that the boundary of the keyhole porosity regime in power-velocity space is sharp and smooth, varying only slightly between the bare plate and powder bed (emphasis ours).” Thus, we believe that the proposed universal scaling is valid for powder bed AM processes although some fitting constants might need to be adjusted to account for uncertainties and biases caused by a powder layer (Mahmoodkhani et al. *Prog. Addit. Manuf.* 4, 109–116 (2019)). This will be an incremental future work.*

Furthermore, we have applied the scaling laws using other data and powder bed machines in the subsection: **Inherent correlation between keyhole and pore formation** of the revised manuscript. The results demonstrate that there is a significant correlation between the proposed Keyhole number and porosity formed during powder bed multitrack and multilayer AM processes. We believe these results demonstrate the effectiveness of the proposed scaling laws in practical AM processing applications and have the potential to be used for quantifying other keyhole-related defects such as spattering and soot.

We added a short discussion in the **Discussion** of the revised manuscript:

“Although the x-ray experiments are conducted on a bare plate in this work, a previous study reports that the keyhole depth does not change significantly by adding a powder layer between 50 μm and 100 μm deep (2). Recently, another study (18) reconfirms this statement by concluding that the boundary of the keyhole porosity regime in process parametric space varies only slightly between the bare plate and powder bed conditions. Thus, we believe that the proposed universal scaling is valid for powder bed AM process although some fitting constants might need to be adjusted to account for uncertainties and biases caused by the powder layer (35). In this study, we apply the scaling laws to a broad set of data and commercial powder bed AM machines. The results demonstrate that there is a significant correlation between the proposed Keyhole number and porosity formed during powder bed multitrack and multilayer AM processes. These results demonstrate the effectiveness of the proposed scaling laws in practice for AM processes, which might have potential to be used for quantifying other keyhole-related defects such as spattering and soot.”

The three references are:

2. R. Cunningham, C. Zhao, N. Parab, C. Kantzos, J. Pauza, K. Fezzaa, T. Sun, A. D. Rollett, Keyhole threshold and morphology in laser melting revealed by ultrahigh-speed x-ray imaging. *Science* 363, 849-852 (2019).
18. C. Zhao, N. D. Parab, X. Li, K. Fezzaa, W. Tan, A. D. Rollett, T. Sun, Critical instability at moving keyhole tip generates porosity in laser melting. *Science*. 370, 1080-1086 (2020).
35. Y. Mahmoodkhani, U. Ali, S. I. Shahabad, A. R. Kasinathan, R. Esmailizadeh, A. Keshavarzkermani, E. Toyserkani, On the measurement of effective powder layer thickness in laser powder-bed fusion additive manufacturing of metals. *Prog. Addit. Manuf.* 4, 109-116 (2019).

4) p.3 and p.4: Can you double-check the exponent of r_0 in equation (1), equation (3), and p.4 line 1?

We have double-checked and fixed some typos in the equation and description.

5) p.3 Figure 1 How did you determine the liquid/solid interface – it is unclear to me based on the contrast of the operando X-ray images. Can you show those images without the dotted line?

Those dashed lines are added manually to highlight the liquid/solid interface, which is observable as a very faint line in the operando x-ray images. A few operando x-ray images with/without dotted line are shown below:

We have added this figure in the Supplementary Information:

Supplementary Fig. 10. Comparison between operando x-ray image and the same image with manually added dashed line at fusion boundary. This dashed line emphasizes the fusion boundary, which is otherwise observable as a faint line that may be obfuscated in print.

6) p.4 line 35: *Based on the supplementary text and equations, the authors said the simulation model is a multi-physics model but line 35 stated that is a ‘...multiphase model...’. Can you verify the claim?*

Our simulation model is a multiphysics and multiphase model. Multiphysics refers to multiple physical phenomena considered in the model such as thermal transport, Marangoni flow, vaporization, and recoil pressure-driven depression in the melt pool. Multiphase refers to multiple states of matter including solid, liquid, and gas. Multiphysics typically covers multiphase; thus, we consistently call our model a “multiphysics model” in revised manuscript.

7) p.5 Figure 2A-D: *Same comment as Figure 1.*

Our response to Comment 5 applies here as well.

8) p.5 Figure 2E-H: *Can you also explain why there are black dots outside the vapour plume area?*

The black dots outside the vapor plume area indicate low-speed flow (<10 m/s), specifically eddies of gas phase induced by the high-speed vapor plume (approximately 200 m/s).

We added a clarification in the caption of Figure 2 in the revised manuscript:

“The black dots outside the vapor plume area indicate low-speed flow (<10 m/s), specifically eddies of gas phase induced by the high-speed vapor plume.”

9) p.5 lines 22 - 25: 'The magnitude of the higher-frequency fluctuation...', it's not clear how you quantify the frequency fluctuation. This should be included in the method section. Can you explain this further?

We used Wavelet Transform (WT) to analyze the normalized power curves and quantify the frequency as it changes over time. The order of magnitude of the dominant frequencies can be approximated from the WT analysis. The WT is implemented in MATLAB 2019 (MathWorks, USA).

We added a short clarification in **Methods: Data processing and quantification** of the revised manuscript:

“Wavelet Transform (WT) (40) was used to analyze the normalized power curves and quantify the frequency as it changes over time. The order of magnitude of the dominant frequencies was approximated from the WT analysis. The WT was implemented in MATLAB 2019 (MathWorks, USA). The time-power curves were converted to time-frequency spectrums. The mother wavelets were the generalized Morse wavelets (41). The sampling frequency used was 1788 kHz.”

Two referenced added are:

40. C. Torrence, G. P. Compo, A practical guide to wavelet analysis. Bull. Amer. Meteor. 79, 61-78 (1998).

41. S. C. Olhede, A. T. Walden, Generalized morse wavelets. IEEE Trans. Signal Process. 50, 2661-2670 (2002).

10) p.5 line 30: Give that the normalized diffusion length is an important parameter to this study, I would recommend that the authors should discuss it in the main text rather than in the supplementary file.

We added a short discussion of normalized diffusion length in the revised manuscript:

“The normalized diffusion length L_d^* is defined as $L_d^* = \frac{\delta_z}{r_0}$ where the δ_z denotes the thermal diffusion length $\delta_z = \sqrt{\frac{\alpha r_0}{V_s}}$. For $L_d^* < 1$ the thermal diffusion length is smaller than the laser beam radius, which means the thermal energy tends to accumulate within the range of the heat source. Such a situation typically results in an elongated melt pool. For $L_d^* > 1$ the thermal diffusion length is larger than the beam radius, and the melt pool shape tends towards circular.”

11) p.6 line 13: Can you clarify the term 'sensible heat melting' as this is not a common term in the field?

The sensible heat of melting is defined as the amount of thermal energy required to change the temperature of a unit volume of a material from ambient to its melting point. The physical dimension of the sensible heat of melting is energy per volume.

We added a clarification in the third paragraph of subsection: **Energy absorption revealed by multiphysics modeling** in the revised manuscript:

“The sensible heat of melting is defined as the amount of thermal energy required to change the temperature of a unit volume of a material from ambient to its melting point. The physical dimension of the sensible heat of melting is energy per volume.”

Further, can you comment on how reflected powers involved in the laser absorption mechanism? ‘Reflected powers’ is not a common term in the field and it was neither discussed in detail within the main text nor the supplementary information.

The reflected power indicates the laser energy loss per unit time due to laser optical reflections. In keyhole mode laser melting the reflected power is approximately proportional to the energy carried by reflected rays escaping from the keyhole. The definition of the reflected power is given by Equation (28) in the Supplementary Information.

*We added a clarification in the third paragraph of subsection: **Energy absorption revealed by multiphysics modeling** in the revised manuscript:*

“The term reflected power describes the laser energy loss per unit time due to laser optical reflections. In keyhole mode laser melting the reflected power is approximately proportional to the energy carried by reflected rays escaping from the keyhole (the definitions of the energy transfer terms are provided in the Supplementary Text).”

12) p.6 line 22: ‘This rationalizes the fact that the vaporization-related parameters, such as boiling point and latent heat of vaporization, are not necessarily included in the scaling laws.’ This is an interesting statement given that the simulation model you used requires the vapourisation parameter. Without the vapourisation parameter, it would have not been possible to confirm that the reflected powers are involving in the laser absorption mechanism. Are the authors ruling out the need for considering laser-vapor plume interaction when it comes to the prediction of the keyhole morphology and defect generation?

We thank the reviewer for pointing this out and agree that a clarification is needed. The vaporization parameters are important for predicting the keyhole morphology and defect generation. We state that the vaporization-related parameters, such as boiling point and latent heat of vaporization, are not necessary to include in the scaling laws for keyhole aspect ratio (Equations 1 and 2). The scaling law for keyhole aspect ratio is derived based on thermal energy balance during the process, and our simulation results reveal that the thermal energy loss due to vaporization is neglectable as compared to absorbed and reflected energies. Thus, in the derivation of the keyhole scaling law we ignore the vaporization-related parameters and focus on heat conduction induced by laser energy absorption. However, we hypothesize that there exists an unrevealed scaling law based on momentum conservation at the keyhole interface, which is driven by vaporization-induced recoil pressure. This scaling law would control keyhole morphology and its evolution.

*We added a clarification in the third paragraph of subsection: **Energy absorption revealed by multiphysics modeling** in the revised manuscript:*

“These results rationalize that we ignore the vaporization-related parameters and mainly consider heat conduction induced by the laser energy absorption in the derivation of keyhole scaling laws (Equations 1 and 2). Because the scaling laws for the keyhole aspect ratio are derived based on thermal energy balance during the process (Supplementary Text), and our simulation results reveal

that the thermal energy loss due to vaporization is neglectable as compared to absorbed and reflected energies. In addition, we hypothesize that there exists an unrevealed scaling law based on momentum conservation at the keyhole interface, which is driven by vaporization-induced recoil pressure. This particular scaling law controls keyhole morphology and its evolution.”

13) p.8, line 10: ‘...existence of a powder bed has only a small effect on the keyhole....’ I think the authors should confirm whether these scaling laws would apply thick powder bed 50 – 150 μm otherwise, this may mislead readers.

We have addressed this comment in the response to comment 3.

14) In the supplementary information, it’s unclear to me how you quantify the length of the melt pool. Can you elaborate on that?

The length of the melt pool was quantified in the x-ray image by measuring the distance between the front end and the tail end of the fusion boundary in the laser scan direction. The length can also be quantified in the simulation results based on the location of the isotherm of the solidus temperature of the material.

We added a clarification in the caption of Supplementary Fig. 6:

“The length of the melt pool was quantified in the x-ray image by measuring the distance between the front end and the tail end of the fusion boundary in the laser scan direction. The length can also be quantified in the simulation results based on the location of the isotherm of the solidus temperature of the material. The depth of the melt pool can be quantified based on a similar metric.”

We thank the reviewer’s constructive comments that significantly improve the quality of the manuscript.

Reviewer 3

1) I believe that what has been achieved here is nothing short of outstanding. The ability to explain how and why keyholing progresses via high-speed synchrotron X-ray imaging is a key outcome of the work. I am skeptical, or rather not as thrilled with the quantification. But the really fun aspect of this paper was reading about the manner in which keyhole formation occurs. Nothing to add or subtract from this paper. I would just suggest to make the diagrams a little "big and barky". Otherwise, nothing to nitpick or complain about.

We thank the reviewer for their positive comments. We have revised some figures to make them clearer and added a discussion targeting the broader research community:

“Although the proposed scaling laws are verified in laser-based AM processes, the methodology and results can be applied to the broader field of energy beam research with some necessary modifications. Possible areas of applicability include laser welding and cladding (32), abrasive waterjet milling (33), and arc/plasma/electron beam-based manufacturing (34). For example, some modifications would be required to apply these laws to electron beam-based AM processes.

We hypothesize two reasons: first, the energy absorption mechanism of electron-material interaction (i.e., electron collision) is different from laser-material interaction (i.e., laser multiple reflection), which could lead to a different form of energy absorptivity in the scaling law. Second, electron beam fusion is typically conducted in a vacuum environment instead of an Argon/Helium shielding environment used in most laser-based AM processes. The vacuum environment alters the vapor plume dynamics and resulting keyhole size and morphology.

This study provides new experimental data obtained using ultrahigh-speed synchrotron x-ray imaging and new simulation data created with a well-tested multiphysics model. These data are particularly useful for the development and validation of statistical and machine learning models. Multiple scaling laws are derived to quantify keyhole aspect ratio and keyhole instability, which is not observable using the traditional post-mortem characterization. A newly developed multiphysics model elucidates energy losses due to various physical mechanisms. We discover an inherent quantitative relation between keyhole instability and porosity formation, which provides better predictive capability as compared to classical qualitative descriptions.”

32. J. Svenungsson, I. Choquet, A. F. Kaplan, Laser welding process—a review of keyhole welding modelling. *Phys. Procedia*. 78, 182-191 (2015).
33. M. C. Kong, S. Anwar, J. Billingham, D. A. Axinte, Mathematical modelling of abrasive waterjet footprints for arbitrarily moving jets: part I—single straight paths. *Int. J. Mach. Tools Manuf.* 53, 58-68 (2012).
34. J. Norrish, *Advanced welding processes*. (Elsevier, 2006).

REVIEWERS' COMMENTS

Reviewer #2 (Remarks to the Author):

The authors have done a great job in addressing all reviewers' comments in the revised manuscript and response letter.

One minor comment: In the final version of the manuscript, the authors should state that the scaling laws described in the manuscript only valid in the quasi-steady state as written in the discussion section.

Reviewer #2

1) The authors have done a great job in addressing all reviewers' comments in the revised manuscript and response letter.

One minor comment: In the final version of the manuscript, the authors should state that the scaling laws described in the manuscript only valid in the quasi-steady state as written in the discussion section.

*We appreciate the reviewer's time and effort in reviewing this manuscript. We have added a short comment to this effect in the **Discussion** of the revised manuscript:*

“The keyhole scaling laws (Equations 1, 2 and 5) are valid for a quasi-steady state melt pool, meaning that the melt pool is well-developed, and the melt pool size and temperature distribution are approximately unchanging in time (although some fluctuations exist in practice due to the highly dynamic multiphase flow).”